# Development of Low Shrinkage Curing Techniques for Unsaturated Polyester and Vinyl Ester Reinforced Composites

**DOI:** 10.3390/ma15092972

**Published:** 2022-04-19

**Authors:** Iqbal Ahmed Moujdin, Husam Saber Totah, Hani Abdulelah Abulkhair, Abdulmohsen Omar Alsaiari, Amer Ahmed Shaiban, Hussam Adnan Organji

**Affiliations:** 1Center of Excellence in Desalination Technology, King Abdulaziz University, P.O. Box 80200, Jeddah 21589, Saudi Arabia; husam.totah@gmail.com (H.S.T.); haboalkhair@kau.edu.sa (H.A.A.); aoalsaiari@kau.edu.sa (A.O.A.); ashaiban@kau.edu.sa (A.A.S.); haorganji@kau.edu.sa (H.A.O.); 2Department of Mechanical Engineering, King Abdulaziz University, P.O. Box 80200, Jeddah 21589, Saudi Arabia

**Keywords:** curing, composite, unsaturated polyester resins, vinyl ester

## Abstract

This work investigated low shrinkage curing techniques and characterization of unsaturated polyester (UPE-8340) and vinyl ester (VE-922) reinforced composite. The reinforced polymeric composite was composed using various amounts (0.1 vol.% to 0.5 vol.%) of methyl ethyl ketone peroxide (MEKP) and the proportion of UPE and VE (5 vol.%) was kept fixed throughout the study. The epoxy matrix was formed using a 3D printed acrylonitrile butadiene styrene (ABS) dumbbell shape mold and the specimen was cured in the presence of air and an inner gas (carbon dioxide) using a customized ambient closed chamber system. The influence of MEKP on UPE and VE reinforce composites was studied by investigating curing kinetics, shrinkage, tensile properties, contact angle, and thermal stability. The CO_2_-cured results show a significant lower shrinkage rate and higher tensile strength and flexural modulus of UPE and VE reinforced composite articles compared with air-cured reinforced composite. These macro-scale results correlate with the air-cured structure, an un-banded smooth surface was observed, and it was found that the lowest amount of MEKP revealed significant improvement in the contact angle of UPET and VE reinforced composites.

## 1. Introduction

Unsaturated polyester (UPE) and vinyl ester (VE) are widely used thermosetting resins due to their low price, light weight, easy processing of the samples, and water resistance. Generally, both thermoset resins are commonly used in coating, composites, laminating, microelectronics, aerospace, civil construction, automotives, and shipbuilding [1,2]. Moreover, very recently, in the water treatment sector, particularly in desalination, their usage has increased tremendously due to the convenient fabrication of commodities components [2]. However, the pure forms of UPE and VE show inadequate thermal and mechanical tendencies. Therefore, the pristine state of these two polyolefin thermoset resins is required to improve their characteristics, particularly the curing and shrinkage issue during the molding process. Compared to VE, the crosslinking of UPE resins is accompanied by a high degree of polymerization [3,4,5,6]. Technically, the UPE and VE reinforced composites undergo residual strains and stresses during the curing cycle [3,4]. These latter effects depend primarily on the polymer matrix’s physical and chemical shrinkage behavior, from the point where curing stresses that become inept are relaxed to any further extent [3]. To overcome the issue of physical shrinkage during the curing process, one of the improvements is to directly impact the tensile strength of the thermoset matrix and fracture characteristics, while by employing keys to increase the acceptance of composites, a competitive and productive fabrication approach is found. Therefore, some studies have been carried out in recent years to develop nonthermal techniques of reinforced polymer composites for the required application, specifically in tailoring desalination commodities components [2,3]. The resins transfer molding, typically called the RTM technique, is an economical approach to creating complex structures with significant quality from thermoset resins [6,7,8].

Besides that, during the curing process, the chemical shrinkage directly affects the structural establishment of thermoset resins, and its characterization is thus of critical concern for demonstrating the ability to regulate residual stresses [4,5] and configure the development of reinforced components [3]. Thus, several investigations have been carried out using various dilatometric instrumental techniques to overcome the chemical shrinkage challenges for investigating the curing-based dimensional changes in thermoset resins. Consequently, the instrumental technique has become a typical technique to measure the volume changes caused by chemical or physical reactions during the curing process. Nawab et al. [3] specifically highlighted the instrumental technique in their detailed reviewed section; these techniques include rheometer [9], pycnometer [9,10], gravimetric method [11,12], fiber Bragg grating [13,14,15,16,17], capillary-type dilatometer [18,19], plunger type dilatometer (PTD) [20,21,22], thermomechanical analyzers (TMA) [23], dynamic mechanical analyzers (DMA) [24,25], and ultrasonic waves for the analysis of residual stress [26], respectively. Since chemical shrinkage equates to the details of the magnitude of cure, perhaps the PTD and differential scanning calorimetry (DSC) technique is the most reliable method to characterize the magnitude of the curing process [22,24,25]. However, except PTD, all the instruments mentioned earlier work in low pressure; this tends to be deleterious for an exclusively precise analysis since the experimental conditions may not be exactly the same [27,28].

Furthermore, several methods have been applied to quantify chemical shrinkage due to the volumetric or dimensional change crosslinking resin. Accordingly, it can be said that there are two typical approaches to measuring resins chemical shrinkages, such as linear (axial) shrinkage and volume dilatometry measurement [29]. Shah and Schubel [30] used several techniques for measuring resin cure shrinkage and evaluated the real-time resin shrinkage, manipulated using a rheometer, pycnometer, and a thermo-mechanical analyzer (TMA) for ambient curing UPE and epoxy resins.

Sadeghinia et al. [31] investigated the cure-dependent properties of a series of epoxy resins with and without filler. Their studies measured the cure shrinkage, the cure kinetics, and coefficient of thermal expansion using DSC and a GNOMIX high-pressure dilatometer. Fernandez-Francos et al. [32] presented a new technique to monitor together the degree of cure and curing shrinkage of thermosetting resins. They used situ ATR FT-IR spectroscopy to achieve quantitative measurements of the degree of cure and shrinkage for thermosets.

As described earlier, the chemical shrinkage is related to the magnitude of resin curing by neglecting the thermal gradients, and shrinkage was linear. However, in fact, the thermal gradients always prevail in the thermoset resins because of the hefty bond among the reaction kinetics, heat transfers, and the low thermal diffusivity of resin, respectively. These variables set off more considerably in the case of molding the bulky parts and may transform the shrinkage curve’s progress a great deal. Therefore, it is essential to quantify these gradients and then determine the chemical shrinkage by considering them.

This study reports comprehensive shrinkage, tensile strength, and flexural modulus properties of UPE and VE reinforced composite using a lower volume of MEKP as a precursor without any accelerator. Furthermore, the behavior of a 3D printed ABS dumbbell-shaped mold for developing UPE and VE reinforced matrix specimens is studied. Both the reactivity and the polymerization shrinkage were recorded during molding under air-curing and CO_2_-curing processing conditions. In addition, we studied the influence of curing temperature at the various molar volumes of MEKP for multiple UPE and VE matrices. We verified the final conversion of the molded samples by measuring the glass transition (*Tg*) temperature and thermal stability via differential scanning calorimetry (DSC). Furthermore, we manually estimated the exothermic reactivity (gel to hardening) as a function of temperature vs. time. DSC confirmed the UPE and VE specimen to understand the molded results.

## 2. Experimental

### 2.1. Materials

The thermoset UPE and VE used in this study consisted of five parts of UPE and VE and 0.1–0.5 parts of hardener (methyl ethyl ketone peroxides: MEKP). Siropol 8340 unsaturated polyester resin (UPE), Hetron 922 Vinyl Ester Resin (VE) and MEKP were kindly supplied by Saudi Industrial Resins Limited, Jeddah, Saudi Arabia. A carbon dioxide (CO_2_) gas of purity 99.9% was used and provided by AHG-Abdullah Hashim Industrial Gases & Equipment Co. Ltd., Jeddah, Saudi Arabia.

#### 2.1.1. Preparation of Dumbbell-Shaped Mold for Reinforce Composite

In this study, MEKP/UPE and MEKP/VE were selected to fabricate reinforced neat composite. Table 1 shows the different portions (vol./vol.) of MEKP used with a fixed amount of UPE and VE. Prior to preparing the dumbbell-shaped reinforced samples (ASTM D638–03), each thermoset (UPE and VE) resin was degassed for at least 30 min in a vacuum oven at 1 bar vacuum pressure. Spring coil mechanical stirring was used to stir the thermoset resins/MEKP mixture at about 500 rpm for at least 2 min or until MEKP completely mixed. Later, the resins mixture was placed for about one minute in an ultrasonic bath to eliminate the trapped air in the mixture.

A dumbbell-shaped ABS mold was designed according to the ASTM D638–03, and 3D printers were used to tailor the dumbbell-shaped mold. Figure 1 shows the 3D printed preform mold size. The bench-scale experimental setup for the air- and CO_2_-cured resin transfer molding system is shown schematically in Figure 2. The preform is set down with a release agent to facilitate the easier removal of reinforced composite samples from the mold later. The mold must be landed on the levelized support. Before spilling the epoxy resins in the dumbbell-shaped mold, a vacuum pump was used to evacuate the air, and then, the resin was infused into the mold. Generally, dealing with the molding process at ambient conditions, a poor merger can lead to the creation of voids and dry spots, which degrade the mechanical properties of the finished product. Thus, after infusing the resin in the preform and resins matrix consolidation, the ambient pressure for the air and CO_2_ curing process must be maintained until the curing phase, solidification, and the dumbbell-shaped samples are ready to be de-molded in the finished sample.

#### 2.1.2. Characterization of UPE and VE Reinforced Matrix Cured under Air and CO_2_

The total shrinkage at ambient temperature (25 °C) of each cured molded dumbbell-shaped sample was measured using a digital vernier caliper and evaluated using the following equation [33],
(1)$$Total\;Shrinkage\;\left(\%\right)=\frac{\sum_{T=25}^{n_{1},n_{2},n_{3}\;}D_{m}-\sum_{T=25}^{n_{1},n_{2},n_{3}}D_{S}\;}{\sum_{T=25}^{n_{1},n_{2},n_{3}}D_{m}\;}\times 100$$
where $n_{1},n_{2},n_{3}$ shows the width, length, and thickness of the mold and specimen, *T* is the temperature (°C), *D_m_* is the 3D dumbbell-shaped mold size before curing the reinforced samples, and *D_s_* is the cured dumbbell-shaped specimen.

#### 2.1.3. DSC Evaluation

DSC studies were carried out following the method described elsewhere [31,32] using Netzsch DSC F3 Maia (Netzsch-Gerätebau GmbH, Selb, Germany). Dry nitrogen gas was purged into the DSC cell with a flow rate of 50. The sample mass was kept at about 10–21 mg in an aluminum hermetic pan. The resin samples were heated from 33 °C to 250 °C at a heating rate of 10 °C/min. After the assessment, the samples were cooled down to room temperature at a similar rate of 10 °C/min. The thermograms were utilized to study the peak max (°C/min), peak height (mW), and glass transition (*Tg*) temperature. The phase transition peak area compared with the calorimetric constant determined for an appropriate standard material allows the phase transition enthalpy to be obtained.

#### 2.1.4. Tensile Strength

The mechanical properties of air-cured and CO_2_-cured dumbbell-shaped molded reinforced composites were measured on an Instron 4411 Universal Tester (Norwood, MA, USA) with a 100 N load cell. All assessments were examined at the ambient condition using a crosshead speed of 50 mm min^−1^. Each sample test was performed three times according to BS EN ISO 527-2:1996.

#### 2.1.5. Flexural Strength Test

To assess the different mechanical properties of the polymer matrix, the transverse rupture strength, also called flexural tests, was also investigated. Typically, this test is used to examine each sample’s stress, strain, and maximum external load. Each test was performed five times according to ASTM D 790 standards. The specimens were tested on an Instron 4411 Universal Tester at a strain rate of 0.10 mm/min and 20 mm/min speed. The force accuracy is 0.5% of applied load and 0.001 mm/min speed resolution. All the samples were performed at ambient conditions. Following are the dimensions of the samples,

total length → 100 mm

span length → 80 mm and 20 mm,

thickness → 5 mm.

Each sample was set aside for three-point bending tests, and the strength of each polymer matrix was calculated using the following relation [34],
(2)$$\sigma_{f}=\frac{3NL}{2wt^{2}}\;$$
where $\sigma_{f}$ shows the flexural strength in MPa, *N* is the maximum load, *L* is the length, *w* is the width, and *t* shows the thickness of the dumbbell-shaped article.

#### 2.1.6. Flexural Modulus Assessment

The flexural modulus assessment of each dumbbell-shaped sample is as distinct as the composite’s competence to deform. Technically, it is calculated from the stress and displacement curve slope as described elsewhere [35]. It is also called a tangent modulus and modulus of elasticity. The following trends are usually used to obtain the flexural modulus [35].
(3)$$E_{f}=\frac{L^{3}m}{4wt^{3}}$$
where $E_{f\;}$shows the flexural modulus, and *m* is the slope of the stress-strain curve.

#### 2.1.7. Contact Angle

To investigate the water absorption tendency of neat UPE and VE reinforced matrix an Attension Theta tensiometer (Gothenburg, Sweden) (instrument was used. The Sessile droplet technique was used to measure each reinforced sample’s contact angle, and each test was performed three times. A fine capillary syringe of 4 µL of a de-ionized water droplet was applied on the selected flat surface of the dumbbell-shaped matrix. Attension image analysis software was used to determine the contact angle. 

## 3. Results and Discussion

### 3.1. Molding and Cured Shrinkage Analysis

The dimensional precision of thermoset epoxy matrix or reinforced composites is a significant issue during manufacturing of the components, particularly when tight forbearances are needed [7]. One of the primary consequences of thermoset resin, particularly the types of early cycles curing resins, is that UPE and VE show dimensional variations in volumetric changes during the cure. Therefore, before producing the required object, we comprehensively investigated the shrinkage characterizations of an air- and CO_2_-cured UPE-8340 and VE-922 matrix. We showed that the set of 3D printed dumbbell-shaped molds made using ABS plastic could be accurately modeled by considering total shrinkage. Before introducing each epoxy mixture, the dumbbell shape ABS mold was lubricated using Dow Corning high vacuum grease. We showed that the warpage of a dumbbell-shaped specimen can be precisely replicated by considering total shrinkage (cure shrinkage + thermal shrinkage). We examined the method of cure shrinkage to minimize it. A decrease in free volume is considered to correlate closely with cure shrinkage. Figure 3 shows the lifted dumbbell-shaped specimen correlation between cure shrinkage and free volume for various combinations of air-cured and CO_2_ epoxy resins and hardeners. The free volume of the epoxy resin correlates with the cured shrinkage of the UPE and VE matrix, and this result supports the above speculation. The overall cured epoxy matrix corresponds to experiments conducted at 26 °C because this is the temperature at which the curing process is reported in the section of curing shrinkage. After the completion of the UPE and VE matrix, we did not observe any physical or chemical effect on the 3D ABS mold (see Figure 3). It is probably because of a high-temperature silicon grease we applied to the mold before pouring the epoxy mixture (epoxy and hardener). As the resin cures, specifically, in a greased layered mold that constrains the environment within the spaces of the dumbbell shape, residual stresses may occur within the resin. These stresses may exceed the depth of both epoxy matrices, which have varying degrees of transformation.

Moreover, the grease layer provides a stress-free ground for the epoxy resins during the curing process.

Usually, based on the degree of crosslinking, the volumetric percentage of cure shrinkage of polyolefins resins is about 7–12% [36,37,38]. Thus, to achieve a low shrinkage profile at ambient conditions, the UPE and VE matrix must select the best ratio of resins and MEKP. We followed a parameter that correlates closely with the cure shrinkage of both resins. We speculated that cure shrinkage reveals a close correlation with the reaction rate of each epoxy group with MEKP because the inter-molecular distance is closely related to the ratio of the chemical reaction between the resins and hardener of each epoxy set. In recent peer-reviewed articles, several researchers have used several predictive cure kinetics approaches [27,33,34,39,40,41,42,43,44,45,46]. For example, Oota and Saka [33] developed a technique to evaluate cure shrinkage and verified that their result of being bent or twisted out of shape is governed by total shrinkage = cure shrinkage + thermal shrinkage. Very recently, Voto et al. [39], carrying out a cure kinetics approach, reported customizing reinforced compositions that focus on matrix constituent reactivity. Perhaps, in this study, we have used a realistic strategy by using Equation (1) to determine the total shrinkage of UPE-8340 and VE-922 at various ratios of MEKP. Thus, Figure 4 and Figure 5 summarized the total shrinkage of two epoxy matrices after curing. The shrinkage results show that in a dumbbell-shaped mold, the upper surface of epoxy matrix cures earlier than that in the overall spine, which agrees with the general experimental result. As shown in Figure 4, with increasing MEKP concentration, the volumetric shrinkage follows a curious trend of uncertainty shrinkage of air-cured and CO_2_-cured UPE-8340 matrices. The lowest (0.1 vol.%) and highest (0.5 vol.%) volume of MEKP showed a higher shrinkage rate in the UPE-8340 matrix. However, the overall CO_2_-curing of the UPE-8340 matrix showed that the shrinking behavior is minimal, ranging from 2.46 percent to 4.31 percent. Based on the volumetric size of dumbbell-shaped cured samples of UPE, they show that a moderate amount of 0.3 and 0.4 portions of MEKP in UPE-8340 provides low shrinkage in air- and CO_2_-cured samples. Nevertheless, in ambient conditions without any external heating and compaction source, probably, the molecular weight is significantly affecting the shrinkage rate of UPE-8340, but this effect is not high; it is just somehow more than 4%.

However, as shown in Figure 5, a linear trend was observed after the curing in the VE-922 matrix; the lower ratios (0.1 and 0.2) of MEKP in the VE-922 matrix show the lowest shrinkage rate, which is about 1.56% to 3.27%. However, in both air- and CO_2_-cured specimens, the shrinkage rate increased once we increased the ratio of MEKP in VE-922. Perhaps, the overall CO_2_-cured shrinkage rate of the VE-922 matrix revealed low shrinkage. Several peer-reviewed articles have reported that air-based gas scavengers greatly influence the specimen’s epoxy matrix 

Moreover, the absence of humidity is also a factor in somehow lesser shrinkage with various portions of MEKP. Thus, the lower shrinkage rate is in good agreement with the CO_2_-cured sample. Compared to VE-922, the shrinkage rate of UPE-8340 shows a higher shrinkage rate, probably due to the molecular weight of UPE-8340, which is higher than VE-822. It is revealed that the exothermic reaction temperature greatly influences the shrinkage of UPE-8340 and VE-922. Many researchers have reported that the higher proportion of hardeners in resins facilitates a significantly higher exothermic reaction and a rapid exothermic reaction leads to a higher curing temperature, usually more than 130 °C [47,48,49,50]. Perhaps high cure temperatures can adversely affect the thermosetting matrix as a result of the high residual stresses and reductions of the percentage of volumetric shrinkage.

Nevertheless, during the experiments we observed that the mismatch in thermal expansion coefficients of each epoxy matrix and the high temperature greased ABS mold contributed to residual stress development. The effects of the chemical reaction of shrinkage development can be seen during the air- and CO_2_-cured period. During the experiments, we found that a higher ratio of MEKP in UPE-8340 and VE-922 created cracking in the specimen. Moreover, the higher proportion of MEKP in each resin faces early curing (about 20 min to 30 min) and higher temperature during an exothermic reaction. Thus, this could also contribute to higher shrinkage in the specimens. Therefore, to overcome the problem of shrinkage in the UPE and VE resins matrix, the third or fourth component says additives are necessary to control the dimensional quality of the epoxy matrix in the mold [37,38].

#### 3.1.1. Effect of Air and CO_2_ Curing on Resin Gel Time and Exotherm 

Figure 6 exhibits the transformation of temperature with time throughout the time of curing resin matrix at different MEKP (hardener) portions. These time-temperature curves reveal the nature of transformation in temperature and the least possible time required for the resin cure with a limited amount of MEKP. As shown in Figure 6, the measured exotherm data show that the use of upper portions (0.5%) of initiator reduced the cure progression time in both the UPE and VE matrix. Similar exotherm graphs were also observed in CO_2_-cured samples for MEKP proportions varying from 0.1 to 0.5 (mL)/5 mL of each resin. In addition, Figure 6 reveals the gel transformation time and additional curing phases such as exotherm peak, peak exotherm time, rate of rising in temperature during curing reaction, and level of cooling of air- and CO_2_-cured specimens. The same figure clearly shows that, for a certain initiator portion, the higher the MEKP part, the lesser the gel time. This fact is principally ascribed to the escalation in the disintegration of peroxides and the creation of highly reactive free radicals that react with monomers to produce crosslinks. Likewise, the gel time variation appears to be fairly low with high levels of MEKP in both thermoset matrices. However, the gel time in the VE matrices is higher than the UPE matrix, and the difference is probably that the molecular chain is longer in vinyl ester resin, which helps the impact of residual stress during the curing process [43]. Moreover, because of the longer chain, the VE resin’s curing temperature is lower than the UPE’s, which could help vinyl ester shrink less during the curing process [43]. Generally, the UPE and VE crosslinking reaction is exothermic. The exothermic reaction temperature can reach about 171–193 °C [44] for a very thick sample. However, such high temperatures were not observed (see Figure 6), which could induce a thermal gradient between the reinforced matrix surface and core [44]. During our experimental studies, we tried the higher amount of MEKP with polymer resins (1:3 vol% to 1:2 vol.%) and experienced a swift curing reaction with a higher exothermic temperature of ~160–170 °C, which almost melted our ABS mold. It is probably due to a higher rate of crosslinking in the dumbbell shape sample.

Middle of the dumbbell shape samples; it can be assumed that distribution of some functional groups among resins and MEKP could occur, leading to the establishment of a concentration gradient in the matrix [44,45,46,47]. Nevertheless, in our future work, we will perform a more precise analysis of the effect of thermal force during polymerization with and without additives in the resins matrix

Comparing air curing and CO_2_ curing in Figure 6, it can be noticed that, even at greater initiator levels, the gel time and the time to exothermic peak of CO_2_ curing demonstrate more significant gel timing and exothermic reaction rate. During the air curing, the UPE and VE matrices experience distinct reactions behavior reliant on their molecular structure and morphology, such as chain networking, chain branching, and monomers crosslinking. Perhaps, during air curing, due to the scavengers, particularly O_2_ and liquid droplets into the air, the monomers crosslink under excess stress and experience degradation or scissoring. Moreover, the air curing of UPE and VE also shows lower gel timing and a higher exothermic reaction rate at various proportions of MEKP probably due to the interaction of free radicals with atmospheric oxygen dissolved in the monomers. Thus, the excessive exotherm peak shows that the oxygen diffusion rate controls more oxidative reactions through the polymer crosslinking.

The CO_2_ curing technique showed the UPE and VE matrix structures, which enhance oxygen diffusion in the monomers and result in more uniform oxidation inside the polymer crosslinking than air curing. In the absence of air, oxygen diffusion is slowed with a lower MEKP ratio (0.1–0.3 vol.%). Nevertheless, under the CO_2_ curing process, we can interpret that the epoxy matrix primarily faces a carbonization process, which probably undergoes oxidative decomposition of the initiator in the first case. In the second case, the reactions generally occur throughout the CO_2_ atmosphere and form a more uniform polymer matrix with minor shrinkage issues. Similar observations were also reported by Rose et al. [48]. Yet, during the curing process, particularly air curing, we observed that in the upper surface of the specimen matrix, the oxygen saturation encourages oxidative degradation. At the same time, the polymer crosslinking occurs in bulk, as shown by the shrinkage of alkyd group concentration, which is associated with the increase of gel portion regardless of various ratios of MEKP.

#### 3.1.2. Differential Scanning Calorimetry (DSC) Analysis

The air- and CO_2_-cured matrix of the studied UPE and VE epoxy was investigated via DSC. Figure 7 and Figure 8 show a typical illustration of the DSC thermograms recorded for air- and CO_2_-cured UPE and VE reinforced matrix at different temperature/min heating rates, in the temperature range ~27–250 °C. The *Tg* was measured during the heating and cooling steps.

In Figure 7, it can be seen that the UPE containing the lowest and moderate amount of MEKP (0.1 and 0.3 vol.%) shows higher *Tg*. Perhaps, UPE containing higher contents of MEKP shows less elevated heat flow peaks, and it has been noticed that as the heating rate rises, the thermal stress become more vigilant on reinforced matrix, and the peak height is also shifted to a higher value. Similar results were also observed in both air- and CO_2_-cured VE matrix resins. Besides that, the effect of 0.1 vol.% of MEKP on both matrix resins is shown in DSC curve not being altered to higher or lower temperatures compared to the higher contents of MEKP in the reinforced matrix [46].

Nevertheless, the results illustrate that UPE and VE reinforced matrices are relatively stable in both curing techniques due to the melting behavior because of variable crystal sizes in the various MEKP systems. The glass transition temperature for both air-cured UPE and CO_2_-cured UPE utilizing lower to greater amounts of MEKP is 136 °C–143 °C for air-cured UPE and 140 °C–146 °C for CO_2_-cured UPE. Figure 7 demonstrates that CO_2_-cured UPE has a greater *Tg* than air-cured UPE, which is owing to the absence of scavengers and likely indicates a substantial mass balance of the oxidation reaction with monomers. Moreover, the UPE containing a moderate amount of 0.3 vol% of MEKP shows the highest *Tg* increment among both curing techniques, which is reveals a very stable crosslinking polymeric network. 

#### 3.1.3. Thermal Stability

DSC data revealed the thermal stability of the air- and CO_2_-cured UPE and VE reinforced matrices with various concentrations of MEKP, which are summarized in Table 2. The visual image of the UPE reinforced matrix shows a stable thermal polymer compared to the VE matrix; perhaps, the matrix that contains a higher portion of MEKP reveals a less durable polymer under the thermal conditions. It can probably be interpreted that there is a higher oxygen content present in both matrixes. Thus, this study shows that the higher portion of MEPK in UPE and VE leads the polymer to a higher volume shrinkage rate and is more degradable under the thermal environment. Thus, in order to prevent UPE and VE instability under high thermal circumstances, it will be essential that the polymer matrix must have an antioxidant component before the curing process. Even though various concentration differences of MEKP used for a reinforced matrix are more, transformation curves derived from the DSC results have already been illustrated in Figure 7 and Figure 8. As the rate of heat increases during the DSC analysis, all of the distinctive temperatures of the curing reaction shift to higher values.

Nevertheless, in Table 2, we have observed no significant difference between the two curing techniques concerning the T (°C), the temperature of the peak maximum (°C/min) in the heat flow curves. In contrast, the reaction between MEKP and epoxy during the curing process begins at shallow temperatures, even below 0 °C. However, in DSC analysis at a lower heating rate of 33–37 °C·min^−1^ the thermal stress on the reinforced matrix seems to be relatively slow at the beginning but accelerates considerably towards the end (high temperatures). As described above, during the curing process, the crosslinking of monomers and initiator involves several different oxidation degradation reactions due to the molar volume of the initiator; thus, probably, the rates of these reactions will vary with temperature in different ways. Thus, a change in the heating rate may favor one reaction over another and change the structure of the crosslinked molecular network. This would explain the decrease in the ultimate *Tg* observed for both systems on increasing the heating rate (Table 1). It can also be seen in Table 1 that the air-cured UPE and VE decrease the final *Tg*. This may be explained either by the scavengers present in the air reducing the crosslink density or by cramming of the reversible inclusion, whose alkyl/aryl chain structure may escalate the crosslinked network’s mobility and influence the curing mechanism as well.

#### 3.1.4. Contact Angle and Water Absorption Comparison of Air- and CO_2_-Cured Specimens 

Understanding the wetting behavior with the exact ratio of MEKP concern is extremely important, as is studying the chemical surface of UPE and VE reinforced matrices. The wettability can also describe the progress of the absorption between the reinforced polymer matrix and applied liquid. The reinforced polymeric components have to depend on the interaction with a specific application, particularly in seawater desalination. Figure 9 and Figure 10 summarize the comparison of deionized water contact angles with their standard deviation values measured for two different cured UPE and VE reinforced matrices. As can be seen, the addition of various concentrations of MEKP into the UPE resins leads to distinct hydrophobic/hydrophilic properties of the obtained specimen. The θ values of the air- and CO_2_-cured UPE matrix showed that the reinforced polymer containing the lowest amount (0.1% and 0.2%) of MEKP represented more hydrophobic behavior, which means there is a lower drag reduction property in water flow application. 

However, in air curing techniques, the lowest content of MEKP in UPE matrix showed contact angles greater than 95, representing an excellent water repellent and hence hindrance properties. The higher contact angle in both reinforced polymeric matrices cured techniques probably revealed lower oxidative degradation due to the lower peroxide content interaction with the monomers. Dirand al el. [44] also reported that during the curing process it is most likely that oxygen is responsible for the creation of such an interfacial hydrophobic smooth layer.

Perhaps, the higher MEKP in the UPE reinforced matrix showed moderate hydrophobicity. Nevertheless, from Figure 9, it is observed that the affinity of the water droplet to the reinforced matrix surface was proportional to the proportion of MEKP in the resin, and the std deviation values show a good agreement of the polymer matrix stability. By adding a higher amount of initiator into the polymer matrix, the contact angle was decreased. At 0.3 wt.%–0.5% MEKP loading, the θ values were decreased by 1.21–1.26-fold, respectively, compared to the 0.1% MEKP-based reinforced composite. At 0.4 vol.% MEKP loading, the contact angle values were decreased sharply by up to 1.26%. Nevertheless, compared to the CO_2_ technique, the air-cured method showed more significant improvement in the contact angles of UPE and VE reinforced matrix, as shown in Figure 9 and Figure 10. It was found in the literature that the contact angle measurement can also provide a level of intercalation between the initiator and the polymer [44,45,46,47]. The water droplet contact angle of VE-922 (see Figure 8) had a lower surface θ than UPE and showed a partial hydrophobic tendency. However, both air and CO_2_ cured.

UPE and VE matrices containing higher amounts of MEKP showed insignificant contact angles lower than 80°. This may be due to the presence of oxygen in double bonds and OH group attachment in aliphatic chains. In addition, probably this could show the imprecise nature of the interfacial interactions of O and OH (typically present in MEKP) which could be inherent in the chemical bonding with polyolefin chain [48]. Moreover, the lower contact angle of the VE matrix in both curing techniques also revealed acid or base formation due to the interactions of the higher molar volume of MEKP. Similar interpretations were also reported by Diran et al. and Abral et al. [44,47].

## 4. Mechanical Properties

### Tensile Strength of UPE and VE

Figure 11 shows the effect of MEKP concentration on the cured samples’ tensile strength. It is observed that a higher proportion up to 0.3 vol.% of MEKP in the reinforced matrix leads to a significant tensile strength in both cured techniques (air and CO_2_). However, the UPE-8340 matrix (cured under air and CO_2_) shows more inconsistent behavior than VE-922; its shows insignificant tensile strength at a higher concentration of MEKP. At a high molar volume ratio of MEKP, the tensile strength of the UPE matrix decreased, probably due to a rapid crosslinking reaction during the curing process and the difficulty of the molecule to be aligned uniformly. It may be due to a higher exothermic reaction temperature during the curing time, which affects the matrix’s tensile strength, or the imbalanced vapor pressure difference between UPE and MEKP; therefore, it has acted (MEKP) as a stress concentrator, which reduces the tensile strength. Safarabadi and Shokrieh [49] and Shokrieh and Ghanei [50] also revealed similar interpretations.

Moreover, the higher molar ratio of MEKP in the UPE matrix probably decreased the polymer chain adjustability resulting from the reduced polymer chain crosslinking in the thermoset structure. On the other hand, the air- and CO_2_-cured VE matrix shows significant behavior and revealed a linear tensile strength curve corresponding to its maximum load due to a higher amount of MEKP. Furthermore, the increased tensile strength of VE in both curing techniques correlates to a crosslinked chemical structure network in contrast to the higher MEKP ratio. Furthermore, the improvement in tensile strength is probably due to an oxidation reaction with the polyolefin structure of vinyl ester. As compared to the air-cured UPE and VE matrix the CO_2_-cured matrix shows significant improvement in the tensile strength and a better load bearing tendency. Nevertheless, the moderate ratio of MEKP in both matrices enhanced the load holding capability, which means 0.3 vol.% of MEKP was a good reinforcing composite for both neat resins.

#### Flexural Properties

To evaluate the bending resistance of air- and CO_2_-cured UPE and VE reinforced matrix composites, each specimen’s flexural strength and modulus were calculated using Equations (2) and (3). Figure 12A,B summarized the flexural strength (A) and flexural modulus (B) properties of each UPE and VE sample. As shown in Figure 12A,B, the flexural strength and flexural modulus of the UPE matrix shows that the sample with 0.3 vol.% of MEKP revealed the highest flexural properties, i.e., 5023.145 MPa (air-cured) and 5127.696 MPa (CO_2_ cured), and the modulus is about 1385.376 MPa (air-cured) and 1489.926 MPa (CO_2_ cured). Figure 12A,B show that increasing the volume percent of MEKP in UPE lowered the flexural properties. The UPE samples cured under CO_2_ revealed better flexural properties than the air-cured UPE sample. The lower flexural properties at higher contents of MEKP is due to a faster curing reaction, which probably caused the rapid crosslinking chain network during the polycondensation polymerization process. On the other side, the VE matrix shows (see Figure 12A,B) a different pattern, i.e., at lower contents of MEKP in the VE matrix, which demonstrates greater flexural strength but a lower flexural modulus. As compared to the UPE matrix, the flexural modulus of air- and CO_2_-cured VE matrix revealed better properties at a lower initiator content. However, unlike the UPE matrix, the 0.3 vol.% content of MEKP in VE corresponded to the lowest flexural properties.

Nevertheless, the flexural strength and flexural modulus of UPE reinforced matrix composites were raised in both curing procedures in the current study at the lowest MEKP concentration in the matrix, although the VE matrix displayed greater flexural strength but lower flexural modulus at lower MEKP contents. The lower contents of MEKP in both resins might form a balanced crosslinking network with monomers, but it also led an extended curing process in both techniques. The extended curing process might be facilitated to reduce the chemical and physical stress during the polymeric chain structure, and the shrinking results are in excellent agreement to validate delaying the curing approach at the presence of molar volume balance between the hardener and resins. Moreover, the hardener and resins’ balance molar volume ratios involve a smooth, free radical chain-growth reaction [8]. Yang and Lee [27] and Merle et al. [8] also reported that at the beginning of the hardener/resins process, the polymerization is initiated once the hardener is mixed with thermoset resins. Principally, the hardener decomposed in the matrix and established the free radicals in the system. At the early stage of polymerization, the free-radicals convert the monomers into gel and simultaneously, for a time, grow and grow as a form of long-chain crosslinking in the styrene monomers system. As the reaction continues, more and more particles are formed until they become crosslinked.

## 5. Conclusions

This work has developed deficient shrinkage curing techniques for UPE and VE reinforced matrix without any accelerator, co-polymer, or additives. The air and CO_2_ closed chamber curing approach has revealed thick interphase in UPE and VE resin cured in contact with a 3D printed ABS dumbbell-shaped mold. In addition, the warpage of a dumbbell-shaped specimen under a closed chamber could be systematically reproduced under an ASTM D638–03 standard, which has revealed deliberating total shrinkage (cure shrinkage + thermal shrinkage).

This interfacial layer decreases from the variation of the relative molar volume of MEKP in VE and UPE resins from the interface to the bulk after curing. The air and CO_2_ curing on resin gel time and exotherm results established that the lower volume of initiator (MEKP) has less impact on chemical and physical stress on the polymer matrix during the curing process.

The results revealed that, for a specific initiator portion, the higher the MEKP part, the lesser the gel time and higher the volume shrinkage. This fact has typically been attributed to the upsurge in the crumble of peroxides and the creation of highly reactive free radicals that react with monomers to produce cross links.

The CO_2_ curing methods have found a better approach for developing low shrinkage specimens with improved tensile and flexural strength than air curing. It is also shown that the flexural modulus of the resin increases from the interface to a depth. Such a phenomenon could play an essential role in the magnitude of the mechanical stress performance of UPE and VE matrices at a lower volume ratio of the initiator.

Nevertheless, only the contact results showed a partial hydrophobic reinforced matrix; this may be due to the attachment of the OH functional group during polymerization. The DSC results revealed that the *Tg* of the CO_2_-cured UPE and VE matrix increased, and the thermal stability results are in good agreement with the CO_2_-cured matrix, respectively. The flexural strength also showed improved results compared to air-cured matrix specimens.

## Figures and Tables

**Figure 1 materials-15-02972-f001:**
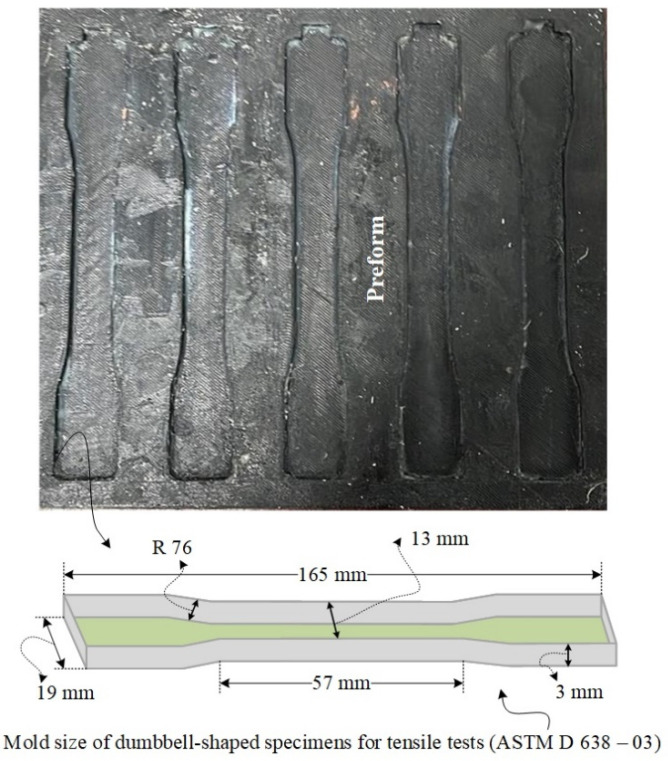
3D printed dumbbell-shaped preform mold.

**Figure 2 materials-15-02972-f002:**
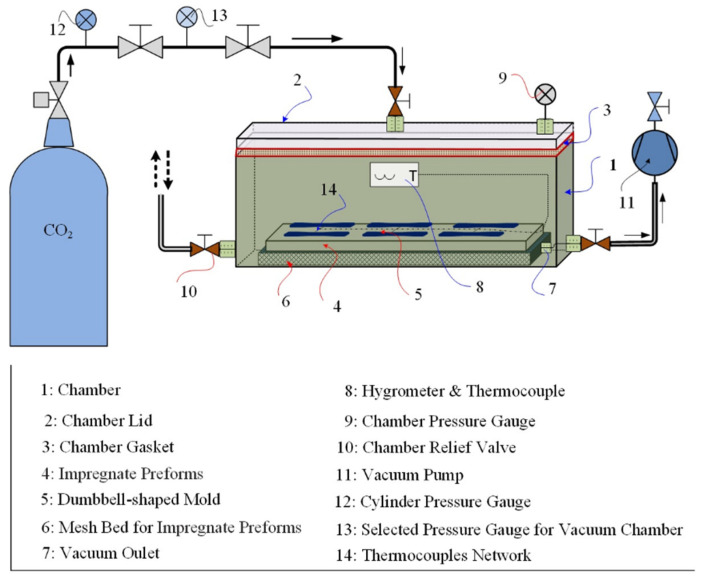
Schematic experimental setup of air- and CO_2_-cured for reinforced composites.

**Figure 3 materials-15-02972-f003:**
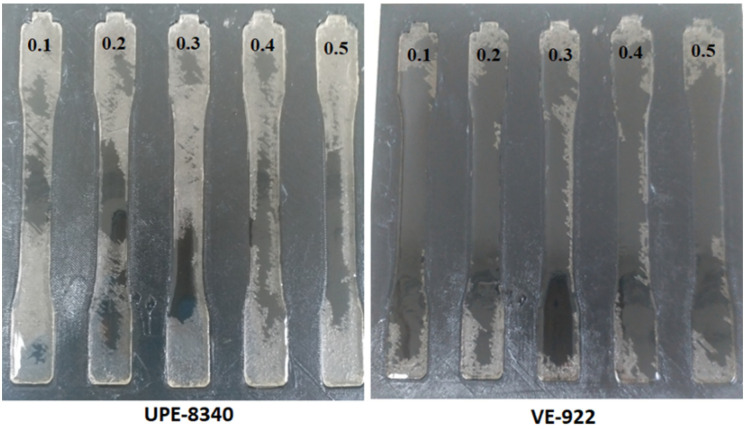
The shrinkage differences (air) between the 3D printed ABS mold and epoxy specimen using various parts of MEKP.

**Figure 4 materials-15-02972-f004:**
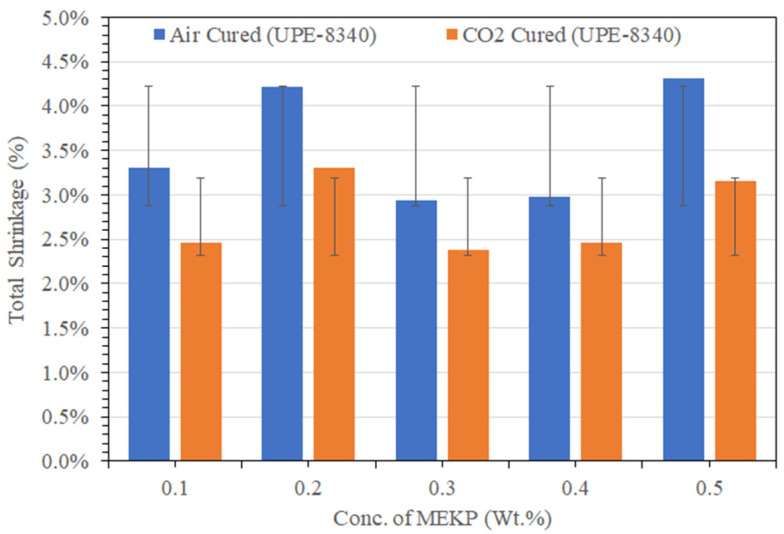
The shrinkage differences between air-cured and CO_2_-cured UPE with various portions of MEKP.

**Figure 5 materials-15-02972-f005:**
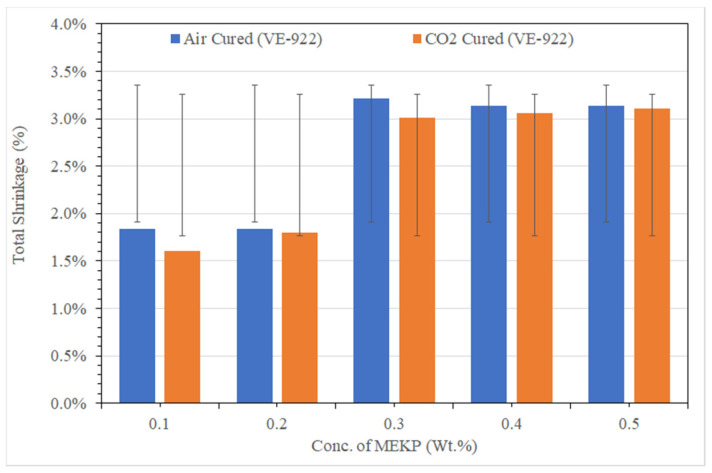
The shrinkage differences between air-cured and CO_2_-cured VE with various portions of MEKP.

**Figure 6 materials-15-02972-f006:**
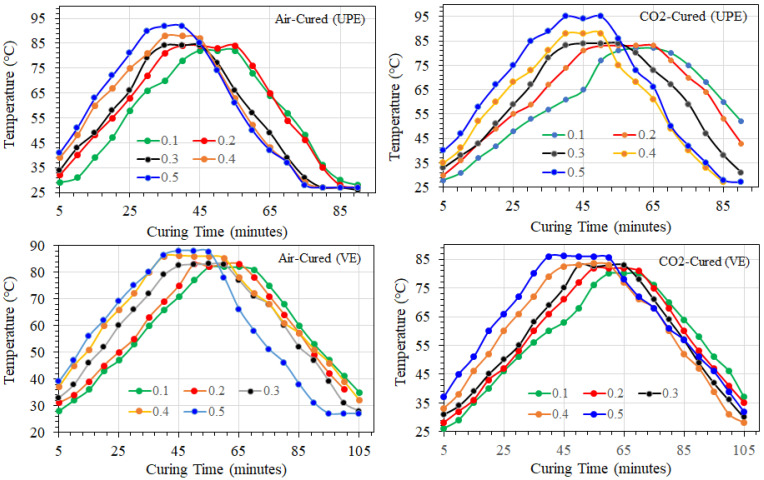
Rate of temperature increments during exotherm reaction as a function of various precursor proportions in UPE and VE matrix system.

**Figure 7 materials-15-02972-f007:**
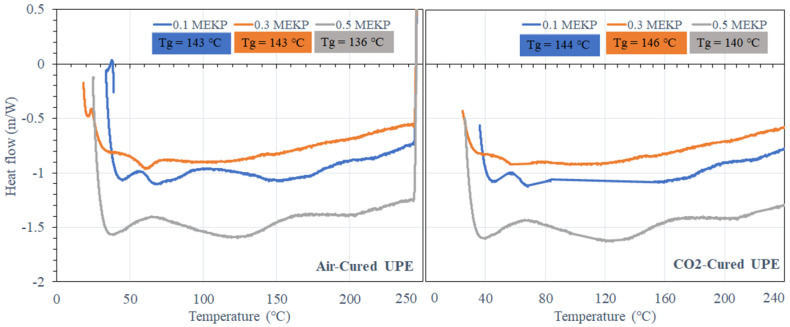
DSC curves of the air- and CO_2_-cured UPE reinforced matrix with various concentrations of MEKP.

**Figure 8 materials-15-02972-f008:**
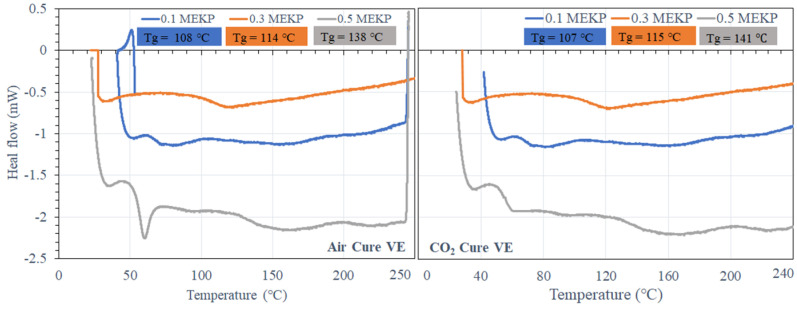
DSC curves of the air- and CO_2_-scured VE reinforced matrix with various concentration of MEKP.

**Figure 9 materials-15-02972-f009:**
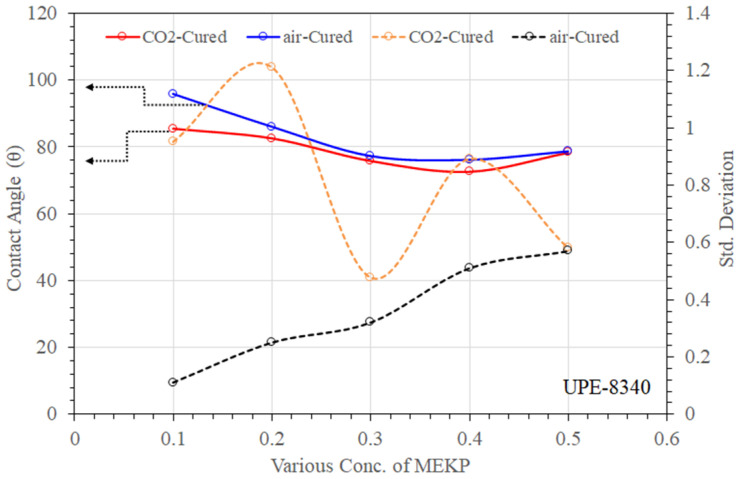
Contact angle and their standard deviations of air- and CO_2_-cured UPE matrix with various amounts of MEKP.

**Figure 10 materials-15-02972-f010:**
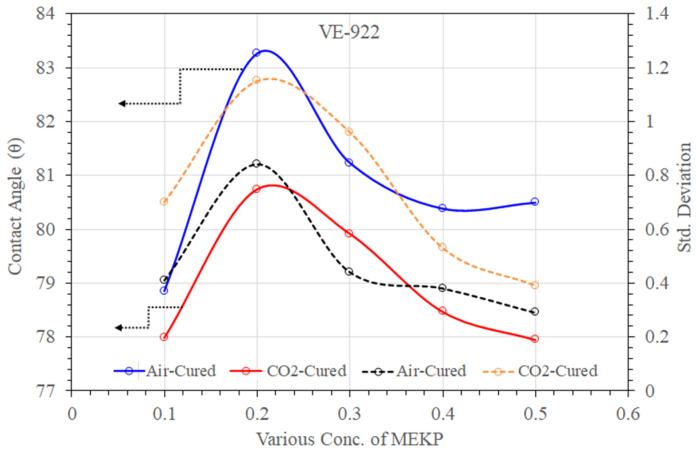
Contact angle and their standard deviations of air- and CO_2_-cured VE matrix with various amounts of MEKP.

**Figure 11 materials-15-02972-f011:**
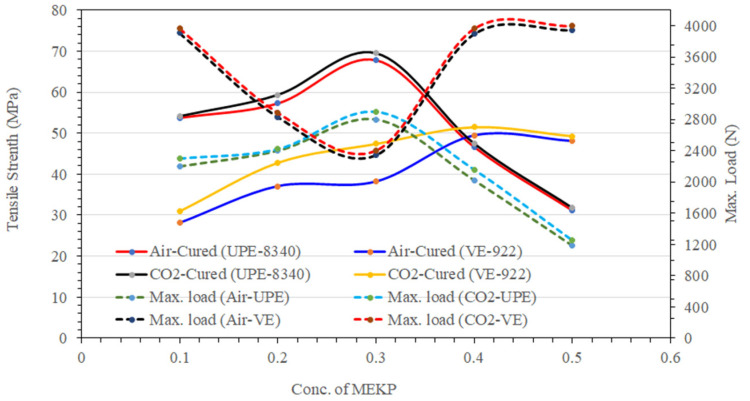
The effect of MEKP concentration (%V/V) on cured samples’ tensile strength.

**Figure 12 materials-15-02972-f012:**
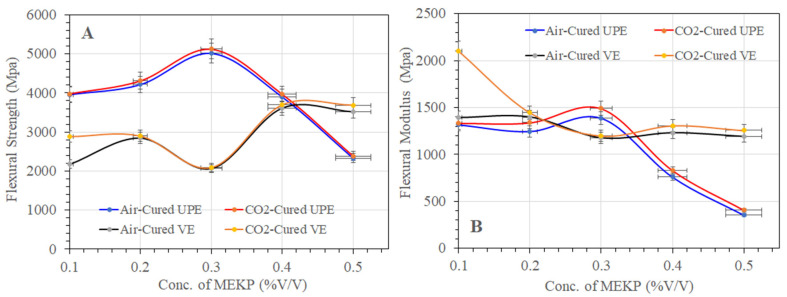
The effect of MEKP concentration (%V/V) on cured samples’ flexural strength (**A**) and flexural modulus (**B**).

**Table 1 materials-15-02972-t001:** Different portions of MEKP used with a fixed amount of UPE and VE.

Composition	A	B	C	D	E
	Proportion (vol.%)				
**MEKP**	0.1	0.2	0.3	0.4	0.5
**UPE and VE**	5	5	5	5	5
**Air**	Ambient condition				
**CO_2_**	99.99%				

**Table 2 materials-15-02972-t002:** Thermal stability of air and CO_2_ reinforced thermoset polymer with various concentrations of MEKP.

MEKP (Conc.)	Air Cured UPE	CO_2_ Cured UPE
0.1	0.3	0.5	0.1	0.3	0.5
	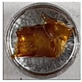	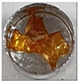	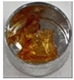	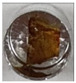	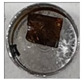	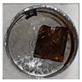
**Heat (J/g)**	2	0.893	1.932	2.4	1.126	2.334
**T (°C)**	57.4–103	49.2–75.5	30–68	57–84	40–80	33–69.7
**Time (min)**	9.3–18.4	7.6–12.9	4.4–11.4	9.1–16.5	6.8–14.2	5.7–11.6
**Peak Max (°C/min)**	69/11.6	60.8/9.9	38/5.4	68.3/12	56.4/10.2	37/6.1
**Peak Hight (mW)**	−0.122	−0.103	−0.232	−0.113	−0.45	−0.195
**Heat (J/g)**	**Air Cured VE**	**CO_2_ Cured VE**
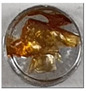	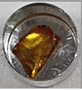	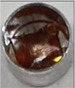	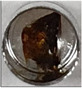	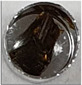	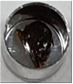
1.705	0.92	1.806	1.851	0.98	1.87
**T (°C)**	62–101	28–66	46.4–72.2	65–110	29–73	51–80
**Time (min)**	10.3–18	3.7–10.9	7–12.2	10.9–19.5	4–13.1	8.1–14.7
**Peak Max (°C/min)**	80.1/13.8	33.4/4.6	60/9.7	80/15.8	36/5.7	60/13.8
**Peak Height (mW)**	−0.101	−0.068	−0.52	−0.105	−0.108	−0.104

## Data Availability

All the research work and data collection were produced at the Center of Excellence in Desalination Technology, King Abdulaziz University, Jeddah.

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
