# Peer review of "Development of Low Shrinkage Curing Techniques for Unsaturated Polyester and Vinyl Ester Reinforced Composites"

_materials, 2022, doi:10.3390/ma15092972_

Round 1

Reviewer 1 Report

The manuscript by these Authors is an interesting piece as regards the preparation of reinforced polymers composites and their characterization. The topic is certainly worthy of investigation and the approach by these authors is correct, at the same time the conclusions they reached are in line with the experimental results they obtained. Anyway, the text suffers from the different problems I highlighted in the attached report. Apart from the suggestions there reported I would suggest also a strong revision of the text in order to correct many typos and mistakes, a revision of the English is also strongly suggested.

Author Response

We have appreciated the understanding of the respected reviewer. Following are replied for the comments,

Agreed,

we have revised the whole manuscript and rectified the typos mistake. The English of overall manuscript has been improved accordingly. All the correction has been heighted in Blue Font.

Reviewer 2 Report

The manuscript entitled "Development of low Shrinkage Curing Techniques for Unsaturated polyester and Vinyl Easters Reinforced Composites" is written well and has significance for the related research field. According to the results, the obtained material overcomes the issue of shrinkage and could be used for used thermosetting resins and other related applications. The manuscript could be considered for publication after a minor revision, particularly: 

1) In the title of the paper, the authors write "Easters". The whole manuscript must be proofread carefully to avoid this type of silly typos and mistakes. 

2) Introduction is well written. It covers the exiting methods and highlights the problem in the related research (shrinkage during the curing process) and proposes the solution ( a comprehensive shrinkage, tensile strength, and flexural modulus properties of UPE and VE reinforced composite using a lower volume of MEKP as a precursor without any accelerator).

3) The methods are sufficient to reproduce the experiments. In the preparation of mold, the schemes are illustrative to show the applied method. In Table 1, the other components (94% of the composite) must be included to make it clear. The other tests on tensile and flexural strength, DSC analysis, and contact angle experiments are sufficiently described. 

4) Some sentences in the Introduction part are vague or not properly linked. For example: "Also used the same device to investigate the epoxy molding compound with and without additives [29]." and "DSC confirmed the UPE and VE specimen to understand the molded results.". These sentences should be re-written (p. 3). 

5) Results and discussion part is written well, however, some sentences need revision. For example, the assumptions should be supported with references: "Several peer-reviewed articles reported that air-based gas scavengers greatly influence the specimen's epoxy matrix". (p.10)

6) The title seems wrong "Figure 5. The shrinkage differences between air cured and CO2 cured of UPE". VE instead of UPE? 

7) The conclusion part must be re-written. The style is as if the authors writing the Results and Discussion part, not Conclusions.  Try to cover the main points as a big picture of the results obtained. 

8) As mentioned before, the manuscript has to be proofread, checked for any grammatical errors and typos. Some sentences must be revised/re-written to avoid confusion. 

9) References are also inconsistent, please check and modify to follow the Materials journal style. 

Author Response

  • In the title of the paper, the authors write "Easters". The whole manuscript must be proofread carefully to avoid this type of silly typos and mistakes.

Agreed, the correction has been mad in the title and as well as rectified all typos mistakes in the manuscript. The corrections has been highlighted in Blue Font in the manuscript.

  • Introduction is well written. It covers the exiting methods and highlights the problem in the related research (shrinkage during the curing process) and proposes the solution ( a comprehensive shrinkage, tensile strength, and flexural modulus properties of UPE and VE reinforced composite using a lower volume of MEKP as a precursor without any accelerator).

Thanks for the understanding of respected reviewer

3) The methods are sufficient to reproduce the experiments. In the preparation of mold, the schemes are illustrative to show the applied method. In Table 1, the other components (94% of the composite) must be included to make it clear. The other tests on tensile and flexural strength, DSC analysis, and contact angle experiments are sufficiently described.

We appreciate the respected reviewer comments related to “other components” perhaps we haven’t used any third component like additives. The reinforced composite of both resins (UPE and VE) was carried out with various MEKP portions. However, we have used CO2 as inner gas for the curing process. Nevertheless, the CO2 with its purity has been included in Table 1. The changes can be found in Blue fonts in the manuscript.

4) Some sentences in the Introduction part are vague or not properly linked. For example: "Also used the same device to investigate the epoxy molding compound with and without additives [29]." and "DSC confirmed the UPE and VE specimen to understand the molded results.". These sentences should be re-written (p. 3).

Agreed,

The sentences have been re-written, and the changes are highlighted in Blue Fonts in the manuscript.

5) Results and discussion part is written well, however, some sentences need revision. For example, the assumptions should be supported with references: "Several peer-reviewed articles reported that air-based gas scavengers greatly influence the specimen's epoxy matrix". (p.10)

Agreed,

We have supported our assumptions with more relevant published work. Following peer-reviewed studies has been placed in the manuscript,

Such as Oota and Saka [33], were developed a technique to evaluate cure shrinkage and verified their result of being bent or twisted out of shape is governed by; total shrinkage = cure shrinkage + thermal shrinkage. Very recently, Voto et al. [39] has carried out a cure kinetics approach was reported to customize reinforced compositions that focus on matrix constituents reactivity.  

6) The title seems wrong "Figure 5. The shrinkage differences between air cured and CO2 cured of UPE". VE instead of UPE?

Agreed,

Figure 5 caption has been rectified and highlighted in Blue Font

7) The conclusion part must be re-written. The style is as if the authors writing the Results and Discussion part, not Conclusions.  Try to cover the main points as a big picture of the results obtained.

Agreed,

We have revised the conclusions and added some of the significance of this study.

8) As mentioned before, the manuscript has to be proofread, checked for any grammatical errors and typos. Some sentences must be revised/re-written to avoid confusion.

Agreed,

We have revised the whole manuscript and rectified typos mistake. The English of overall manuscript has been improved accordingly. All the correction has been heighted in Blue Font.

9) References are also inconsistent, please check and modify to follow the Materials journal style.

Agreed,

We have revised the reference accordingly and added a few more relevant references (can be seen in Blue Fonts). Moreover, the reference list has been formatted according to the instructions of the Materials.

Round 2

Reviewer 1 Report

The manuscript has been improved but my suggestions were only partially applied in the manuscript.

This manuscript is a resubmission of an earlier submission. The following is a list of the peer review reports and author responses from that submission.

Round 1

Reviewer 1 Report

In this contribution, Totah et al. processed low-shrink composite using unsaturated polyester and vinyl esters in presence of precursor. Based on the presented data and explanation, I don't feel that the manuscript is suitable to publish in Polymers. I feel that the authors summarized the observations and did not address necessary meaningful explanations which weaken the scientific insight of this study. The work doesn't present much insight to a new discovery in this field. One most important observation is that the manuscript needs very careful reading and thorough revision. There are a lot of error (spelling, sentence construction, typo, and grammar) in the text which makes it hard to follow the manuscript. I would recommend submitting this work in another journal.

My major concerns are listed below:

-  Authors listed the steps performed in this work at the end of Introduction section. However, a clear statement on the novelty of the research is missing. Please state what is new in this study.

- Figure 1 shows the dumbbell-shaped ABS mold which is produced following standard ASTM D638 – 03. Figure 1 should be removed or presented in the Supplementary information. Authors have used standard ASTM D638 – 03 and the figure is not adding any new information on top of standard sample preparation procedure.

- There are several standard equations presented in this manuscript. These could be presented in the Supplementary section or removed from the manuscript. 

- Table 1 is confusing. Why is ratio presented with a unit? In the later part of the manuscript, the samples were named as UPE-8340 and VE-922 and these should be introduced in this table.

- “In Figure 3, it is observed that there is no physical or chemical effect on the 3D ABS mold.” – this is not clear at all from the figure. What does it mean by the physical or chemical effect? Please clarify this statement in the text.

- “… because the inter-molecular distance is closely related to the chemical reaction ratio between resins and hardener of each epoxy set.” – what is a chemical reaction ratio?

- “Thus, Figure 4 and Figure 5 summarized the total shrinkage of two epoxy matrix after curing.” – Please use total shrinkage (%).

- “the molecular weight is greatly affecting the shrinkage rate of UPE-8340 but this effect is not high it’s just somehow more than 4%.” – This statement is confusing. Please clarify.

- “Several peer-reviewed articles reported that air-based gas scavengers greatly influence the specimen's epoxy matrix curing and shrinkage.” – Please provide references.

- Please confirm the error bars presented in figure 4 and 5.

- “In both Figure 7, it can be seen that the UPE contains the lowest and moderate amount of MEKP (0.1 and 0.3vol.%) shows higher Tg perhaps UPE contains higher contents of MEKP shows more elevated exothermic peaks, and it has been noticed that as the heating rate increases, the area of the exothermic peak increases, and the height is also shifted to a higher value.” – This explanation on DSC spectra needs revision and further clarification.

- Explanations presented in the section 3.5 are not clear as well. There are random statements, and the arguments are not connected to each other. Rewriting of the entire section is recommended.  

Reviewer 2 Report

There are serious flaws in the paper. Most of it are addressed in the attached document. Generally speaking, the results presented in figures are not consistent with the results presented in the text. There are a lot od speculations.

The English requires extensive editing. 
